# Adaptation mechanism of the adult zebrafish respiratory organ to endurance training

**Matthias Messerli**[1], **Dea Aaldijk**[1], **David Haberthür**[1], **Helena Röss**[1],
**Carolina García-Poyatos**[2], **Marcos Sande-Melón**[2], **Oleksiy-Zakhar Khoma**[1], **Fluri A. M. Wieland**[1], **Sarya Fark**[1], **Valentin Djonov**[1]*

**1** Topographic and clinical Anatomy, Institute of Anatomy, University of Bern, 3012 Bern, Switzerland,
**2** Developmental Biology and Regeneration, Institute of Anatomy, University of Bern, 3012 Bern, Switzerland

☸ These authors contributed equally to this work.
\* djonov@ana.unibe.ch

**Data Availability Statement:** The tomographic data of the delineated gills are available on the Open Science Framework under https://osf.io/a5esx/. The full analysis is available as a Jupyter

## Abstract

In order to study the adaptation scope of the fish respiratory organ and the $O_2$ metabolism due to endurance training, we subjected adult zebrafish (*Danio rerio*) to endurance exercise for 5 weeks. After the training period, the swimmer group showed a significant increase in swimming performance, body weight and length. In scanning electron microscopy of the gills, the average length of centrally located primary filaments appeared significantly longer in the swimmer than in the non-trained control group (+6.1%, 1639 $\mu$m vs. 1545 $\mu$m, p = 0.00043) and the average number of secondary filaments increased significantly (+7.7%, 49.27 vs. 45.73, p = 9e-09). Micro-computed tomography indicated a significant increase in the gill volume (p = 0.048) by 11.8% from 0.490 mm$^3$ to 0.549 mm$^3$. The space-filling complexity dropped significantly (p = 0.0088) by 8.2% from 38.8% to 35.9%., i.e. making the gills of the swimmers less compact. Respirometry after 5 weeks showed a significantly higher oxygen consumption (+30.4%, p = 0.0081) of trained fish during exercise compared to controls. Scanning electron microscopy revealed different stages of new secondary filament budding, which happened at the tip of the primary lamellae. Using BrdU we could confirm that the growth of the secondary filaments took place mainly in the distal half and the tip and for primary filaments mainly at the tip. We conclude that the zebrafish respiratory organ—unlike the mammalian lung—has a high plasticity, and after endurance training increases its volume and changes its structure in order to facilitate $O_2$ uptake.

## Introduction

Endurance exercise is widely known for its beneficial effects on health due to the physiological changes it promotes in the cardiovascular system of vertebrates. The cardiovascular system itself forms part of the pathway of oxygen, which can be divided into four different elements: gas exchange organ, heart and blood, microvasculature, and mitochondria [1]. Taken together, adaptive responses within the pathway of oxygen lead to an increase in the total oxidative capacity, expressed as VO$_2$ max (maximal oxygen consumption, L/min). In humans, for

notebook on https://github.com/habi/zebra-fish-gills and via Zenodo at http://doi.org/10.5281/zenodo.3342451.

**Funding:** The present work was supported by the Swiss National Foundation (31003A_176038) and Swiss Cancer Research (KFS-4281-08-2017) to MM. The funders had no role in study design, data collection and analysis, decision to publish, or preparation of the manuscript.

**Competing interests:** The authors have declared that no competing interests exist.

example, a highly trained long-distance running athlete may be able to achieve a 2-fold difference in $VO_2$ max compared to untrained individuals [1]. It was shown that (in untrained humans) the limits of adaptation to endurance exercise are due to the oxygen-transporting (supply) factors rather than to the mitochondrial function (demand factor) [2, 3]. Each of the four parts of the pathway has been studied separately for training-induced modification possibilities in various species. In order to understand the mechanisms leading to those changes, let us have a closer look at the different steps of the respiratory cascade, starting with the final point of respiration, the mitochondria.

It is known that (aerobic) exercise provides a strong stimulus for adaptations at the level of mitochondria. A review of cross-sectional and training studies with humans confirmed that physical activity increased both mitochondrial function (as determined by mitochondrial respiration) and content (assessed by citrate synthase activity) [4]. This also applies for other vertebrates, such as fish. A significant increase in the mitochondrial density in red muscle fibres has been shown in zebrafish larvae after training (27% in swimmers versus 21% in the control group) [5]. In adult zebrafish, the number of mitochondria seemed to increase as well, as indicated by real-time PCR of genes encoding mitochondrial enzymes [6].

Regarding the level of microcirculation, endurance training has proven to facilitate capillary proliferation in skeletal muscles of different species. For example, 40 min on a bicycle ergometer for 8 weeks at an intensity of 80% $VO_2$ max led to a 20% increase in capillary density in human quadriceps femoris [7]. In mice, undergoing a voluntary running wheel training for 6 weeks, capillary-to-fibre ratio (CF) in the plantaris muscle was significantly higher (+76%) compared to their untrained littermates. Additionally, they showed an increase in capillary tortuosity (+16.3%) and a reduction of the pericapillary basement membrane thickness (-16.5%) [8]. Also in fish (cyprinids) undergoing endurance training, an increase in CF-ratio in red muscle fibres has been observed [9]. Microcirculation in fast skeletal muscle of zebrafish (*Danio rerio*) is altered through training as well (+54% of capillary density, relative to non-exercised fish) [10].

It is well-established that human athletes have a considerably higher blood volume and a higher total haemoglobin than untrained individuals, as reported by Heinicke et al. [11]. A recent study with 9 previously untrained adults undergoing 8 weeks of cycle ergometer training (60 min, 3-4 times/week) showed a 12% increase of red blood cell volume and a 13% increase in blood volume with a slight decrease of haematocrit (-3%). The authors explained this with the expansion of plasma volume happening at the same time [12]. A meta-analysis on echocardiographic reports of athlete hearts showed a larger left ventricular internal diameter (end-diastolic) with an increased ratio between wall thickness and internal diameter (+7.8%) of male competitive long-distance runners compared to matched controls. Furthermore, the calculated left ventricular mass was 48% larger [13]. In fish, the cardiovascular system responds in a similar way with heart hypertrophy relatively to body weight and increased haemoglobin levels (shown in *Salmo gairdneri, Walbaum*) [14]. In zebrafish, Rovira et al. could show a significant increase in the ventricular area normalised by body weight and a significant increase in the number of proliferating cardiomyocytes, suggesting a hyperplastic adaptation of the heart [15].

Adaptive changes in the lungs, however, remain much more controversial than the rest of the pathway of oxygen. A longitudinal 3-year follow-up study with 453 children, aged 8-16 years, showed that the forced vital capacity (FVC) was higher in swimmers compared to tennis players and gymnasts and in all three athletic groups, FVC was higher than in non-athletes [16]. Armour et al. compared elite male swimmers and elite male long-distance athletes with a non-athlete control group. Several respiratory parameters were significantly increased in swimmers compared to runners and controls, namely total lung capacity, vital capacity,

inspiratory capacity, forced expiratory volume in one second (FEV1) and pulmonary diffusing capacity. As the swimmers showed the same alveolar distensibility as runners and controls, the authors concluded that they had achieved greater lung volumes by developing wider chests and that more likely the number of alveoli was increased instead of their size [17]. Another study compared 16 preadolescent girls, five of which underwent 1 year of intensive swimming training (12 hours per week). A control group of 11 girls participated in various sport activities for 2 hours per week. Before the training period, there were no significant differences between the two groups for vital capacity, total lung capacity, functional residual capacity and FEV1. After 1 year the swimmers had significantly higher values for the studied parameters than the controls, with a similar physical development [18]. These results stand in contrast with the results of a recent study, in which 11 female swimmers and 10 controls (age 11—14 years) were examined before and after one season of competitive swimming. Swimmers had a greater total lung capacity and peak expiratory flow rate than controls but these parameters did not improve significantly after the training season. Therefore, the study suggested that competitive swimming did not affect lung growth during puberty and that large lungs of swimmers are inherent rather than induced [19].

So, not only is there an ongoing discussion about the adaptive possibilities of the respiratory organ, but it also remains unclear whether an increased respiratory volume would be achieved by wider alveoli or an increased number of alveoli. This is an important difference since an increased alveolar number would notably enhance the gas exchange surface, while an increased alveolar diameter would have a much smaller effect. However, the data on exercise-induced lung adaptation in humans mainly come from children and adolescents, when the lungs are still very plastic. In a study with lungs from human autopsies at the age of 2 months to 15 years, the number of alveoli increased exponentially from birth to the age of two, with continuing growth at a reduced rate throughout adolescence [20]. The current opinion on healthy adult lungs is that they do not grow anymore. No compensatory lung growth could be measured in patients with lung cancer within a 1 year of follow-up after pneumonectomy or lobectomy [21]. Nevertheless, there is a case report showing the proliferative capacity in an adult after loss of lung tissue. Lung regrowth including an increase in the number of alveoli was shown in a 33-year-old woman during the 15 years following a right-sided pneumonectomy [22]. In rodents, lung regrowth after pneumonectomy has already been shown in different studies. For example, Voswinckel et al. could show a complete restoration of the initial lung volume (calculated from stereological quantification) after resection of the left lung within 21 days after surgery [23]. Another study, also with mice, documented lung regeneration after pneumonectomy *in vivo*. They could show a regrowth of alveoli in the remaining right lung and an increase in the alveolar surface area, which achieved 100% of the values of both lungs before surgery [24]. The notable difference in the studies with rodents compared to the human case report is the time it takes for adaptation: whereas mice can restore their lung volume within a month, in humans it is rather a process of years. This might be a reason why human lung adaptation possibilities have not been verified to this day.

To summarise, the different parts of the pathway of oxygen show similar training-induced adaptation mechanisms in diverse vertebrates including humans, but no data about the morphological response of the respiratory organ to training have been published yet. For this pilot study, zebrafish were chosen as a well-established model in exercise physiology and in respect to the '3R principle' of animal experimentation. As gas exchange and transfer in fish is less efficient than in vertebrates, oxygen uptake might be a bottle-neck in the pathway of oxygen. Additionally, the anatomy of the gills allows further growth within the opercular chamber, whereas the plasticity of adult human lung might be limited due to the rigidness of the chest

restricting its expansion. Our working hypothesis was that exercise would increase oxygen consumption and improve oxygen uptake via gill enlargement.

## Materials and methods

### Animals

For this study, a total of 3 training cycles with 3 fish groups were performed (total number of fish: 52, equal numbers of males and females). For each training cycle, Tg(*fli1a*:eGFP)$^{y7}$ zebra-fish [25] at the age of 18 to 24 months were randomly divided into two groups of 10 fish (6 fish for the third experiment round), a control group and a swimming group. Embryos for this fish line were originally obtained from the Zebrafish International Resource Center. The choice of transgenic line was made because this study was part of a larger project on exercise-induced angiogenesis. All fish were kept in a conventional fish facility at the Institute of Anatomy of the University of Bern with a diet of brine shrimp (*Artemia*) and dry fish food (Gemma Micro 300, Skretting, USA) twice a day. The fish housing system was from Tecniplast, Italy, with 10 fish per 3 l tank and in a gentle water flow. Water parameters were 25°C, conductivity 500 $\mu$S, pH 7.4 with a 12 hours/12 hours day-night cycle. During the time of the experiment, they were kept in their respective groups and separated only for speed assessments and respirometry. Euthanasia of all fish of each training cycle (swimmer and control) took place at the same day, immersing them in 0.250 mg/ml buffered tricaine methanesulfonate (Sigma-Aldrich, Inc.) in order to anaesthetise the fish before decapitation. The animal experiments conformed to the guidelines of the Swiss government and were prospectively approved by the Bernese cantonal veterinary office under the licenses BE59/15 and BE45/19. There is no Institutional Animal Care and Use Committee (IACUC) that pre-approves experiments at the University of Bern or other relevant ethics board needed for our experiments.

### Endurance training and speed assessment

Training took place in a swim tunnel respirometer (Model SW10100 from Loligo Systems, Viborg, Denmark, chamber volume 10 l, test section 40 × 10 × 10 cm) with a laminar water flow. At the beginning of the training period, a speed test was performed. Fish started to swim at a speed of 0 cm/s with an increase of 5 cm/s every two minutes. Maximum speed was defined as the moment when a fish was not able to swim anymore but landed in the mesh at the back end of the water channel. Both swimming speed at the end and total performing time were documented. The average of the maximum speed of the 10 fish in the training group was calculated and 65% of this value defined as the training speed. After a second speed test at the end of the third week of the training period, the training speed was increased for the last 2 weeks according to the training effect of the first weeks. Control fish stayed in their regular tanks.

The training protocol was established according to Palstra et al. with the exception that during the speed test we increased the speed every 2 minutes instead of every 10 minutes [26]. Fish underwent a 6-hour training (10 am to 4 pm) 5 days/week for a total of 5 weeks. As we observed in the first two exercise rounds that the increase in performance happens mainly in the first 3 weeks, we shortened the training period from initially 5 to now 3 weeks of training for the third group. All the other parameters like training hours per day and speed stayed the same. During the whole training period, the fish were kept in the swim tunnel day and night. After the last swim assessment fish were kept individually for the final respirometry until sacrifice.

The weight of the fish was taken from living fish (only from males), measuring the weight of a water tank without and with fish inside. The length of the fish was measured after anaesthesia with tricaine from the head to the tail fin (excluding the fin) with a precision of 0.5 mm.

## Respirometry

In order to be able to measure the oxygen consumption of the fish both in rest and during exercise, we used another swim tunnel respirometer (Model SW10000 from Loligo Systems with a chamber volume of 170 ml), a Witrox 4 oxygen meter (Loligo Systems) with an oxygen-sensitive optode and a temperature sensor (DAQ-M instrument, Loligo Systems). This allows to measure the oxygen content in the water and therefore the oxygen consumption of the fish, while they stay in the swim tunnel. The data were processed by the AutoResp version 2 Software (Loligo Systems). As the system is very sensitive, water temperature and environment such as noise, light, and movement should stay as constant as possible. The temperature in our respirometry tunnel was kept at 25˚C, like in the whole facility. Fish were left in the tunnel for 15-20 minutes to get familiar with it and reach a basal state of oxygen consumption. During this time the water was constantly exchanged in order to have oxygen saturated water. Afterwards, the system was closed and the decrease of oxygen in the water was measured while the fish were swimming at a moderate speed (training speed).

The electrode was calibrated using water from the system of the fish facility and defining this as 100% saturation. 0% saturation was defined as the saturation when the electrode was put in sodium hydrosulfite (dithionite) $Na_2S_2O_4$ solution (Sigma-Aldrich, Inc.).

## Fixation and critical point drying

Both samples for scanning electron microscopy (SEM) and for micro-computed tomography (micro-CT) were critical point dried. Fixation of samples for SEM was done with Karnovsky fixative (2.5% glutaraldehyde (25% EM grade, Agar Scientific Ltd.) + 2% paraformaldehyde (PFA, Merck) in 0.1 M sodium cacodylate (Merck), pH 7.4), in which the samples were kept until critical point drying. Whole heads with gills *in situ* for micro-CT were fixed in 4% PFA for 2 days. Critical point drying was performed using an ascending alcohol series (70%—80% —96%) for 15 min each for dehydration. Subsequently, the samples were immersed in 100% ethanol for $3 \times 10$ min and then critical point dried in an Automated Critical Point Dryer Leica EM CPD300 (Leica Microsystems, Vienna, Austria) with 18 steps of ethanol—$CO_2$ exchange.

## Scanning electron microscopy (SEM)

For scanning electron microscopy, critical point dried gills were sputter-coated with 10 nm of gold with a sputter coater (Oerlikon Balzers, Liechtenstein). Subsequently, the gill arches 2 and 3 of 20 fish (10 controls and 10 swimmers) were scanned with a scanning electron microscope Philips XL30 FEG (Philips Eindhoven, Netherlands) at a magnification of $38 \times$ and a beam accelerating voltage of 10.0 kV. From those images, the length of the longest 5 primary filaments at the centre of each arch was measured (n = 50 for each group). Secondary filaments on the respective primary filaments were counted. The choice to define a region of interest was taken due to the massive amount of both kinds of filaments within the whole gill organ. Measurements were done in Fiji [27].

In order to document a possible mechanism of growth at the tip of the gills, samples were additionally scanned by a Quanta SEM (FEG 250, Thermo Fisher) at a magnification of 5000×, a beam accelerating voltage of 20 kV and a pixel dwell time of 10 $\mu$s. Pictures of different swimmers were taken in order to exemplarily show different stages of growth.

## Micro-computed tomography (micro-CT)

After critical point drying, as described above, the heads of 20 fishes (10 swimmers and 10 controls) were imaged on a Bruker SkyScan 1172 high-resolution microtomography machine (Bruker microCT, Kontich, Belgium). The X-ray source was set to a voltage of 50 kV and a current of 167 $\mu$A. For most of the dried fish heads, we recorded a set of 3979 projections of $4000 \times 2672$ pixels at every 0.05˚ over a 180˚ sample rotation. For some of the heads, we used a so-called wide scan where two projection images are stitched side-to-side making the projection images approximately two times larger laterally; this was necessary if the fish head was not able to be fitted into the field of view of a single camera window. Every projection was exposed for 890-2005 ms (depending on the sample), six projections were averaged to one to greatly reduce noise. This resulted in scan times between 6 and 19 hours and an isometric voxel size of 1.65 $\mu$m in the final data sets. The projection images were then subsequently reconstructed into a 3D stack of images with NRecon (Bruker, Version: 1.7.0.4).

After reconstruction, we manually delineated the gills in CT-Analyser (Bruker, Version 1.17.7.2+) and exported these volumes of interest (VOI) as a set of PNG images for each fish head. These sets of images were then analysed with a Python script in a Jupyter notebook [28]. The full analysis is freely made available on GitHub [29].

Briefly, we used a simple Otsu threshold [30] to binarize each VOI image into gills and background. The gill volume was then simply calculated as the volume of all the binarized pixels. The organ area was extrapolated with two-dimensional binary closing of the thresholded gill image and summation of this image.

## Immunostaining and confocal imaging

After fixation in PFA 4% for 4 hours, gills were washed $3 \times 20$ min with phosphate-buffered saline (PBS) and then immunostained with anti-bromodeoxyuridine antibody (BrdU, Sigma-Aldrich, Inc.). Therefore, during the training period, the fish were exposed to BrdU in the water once a week overnight for a total of 3 times (on non-swimming days). The concentration of the BrdU was 2 mg/ml (diluted in E3 medium). 3 days after the last exposition, fish were sacrificed, gills were extracted and fixed as mentioned above. Before the staining procedure, tissue clearing was achieved with Cubic I solution for 3 days (Urea, N,N,N',N'-Tetrakis (2-Hydroxypropyl)ethylenediamine 98%, Triton X-100, in $H_2O$ dest. [31], all reagents from Sigma-Aldrich, Inc.). Staining was performed with an anti-BrdU primary antibody (mouse, BD Pharmingen) in a dilution of 1:150 in 5% bovine serum albumin (BSA, Sigma-Aldrich, Inc.) for 72 hours, and as a secondary antibody the Alexa Fluor 568 (anti-mouse, Thermo Fisher Scientific) at a dilution of 1:250 for 60 hours was used. Simultaneously, the staining of the endothelium was performed with anti-GFP (rabbit, Aves Labs, inc.) and Alexa Fluor 488 (anti-rabbit, Thermo Fisher Scientific) as a secondary antibody. Counterstaining of the nuclei was done with DAPI (Invitrogen) during the last night of incubation with the secondary antibody.

For quantification of mitoses, confocal images were acquired with a LSM Zeiss 880 with a $40 \times$ W/1.1 objective. From randomly selected tips of primary filaments of gill arch 2 and 3, volumes of interest of 212 $\mu$m $\times$ 212 $\mu$m $\times$ 50 $\mu$m, with pixel size 0.21 $\mu$m in xy, 1 $\mu$m in z, were scanned (4 images per fish, 6 swimmers and 6 controls, total of 24 images per group). Quantification of the mitoses was done as batch analysis in Imaris version 8.4. (Bitplane, USA) in the following way: DAPI or BrdU positive nuclei were identified as spots of 7 $\mu$m diameter (14 $\mu$m in z), and data were filtered by quality ($>5$), by number of voxels ($>8200$), and by median intensity in the respective channel (20-160). These criteria correctly identified nuclei in most samples, leaving out false-positives (antibody precipitates etc.) or false-negatives. Despite fine-

tuning of the parameters, the criteria were never optimal for all samples due to the sample heterogeneity. However, we verified that even if these criteria were altered, the ratios of spot count between swimmer and control group stayed unaffected.

## Statistical analysis

The full statistical analysis can be found in the analysis notebook [29]. Briefly, for each parameter described below, we tested with a Shapiro-Wilk-test [32] if the data is not significantly differing from a normal distribution. A Levene-test [33] showed us that the data is very probably not normally distributed. A Kolmogorov-Smirnov test [34] showed us that the data has equal variance, enabling us to use a one-tailed Student's t-test which assumes equal population variances [35]. All numerical values in the text are given as averages ± standard deviation. The violin plots are limited to the range of the observed data, the inner lines show the quartiles of the respective distributions. P-values in the text and figure legends are given as precise numbers, as suggested by Amrhein et al. [36]. In the plots, we denote p-values smaller than 0.05 with *, p-values smaller than 0.01 with ** and p-values smaller than 0.001 with ***.

## Results

### Swimming activity, body size and behaviour

Similarly to previous studies, the measured swimming performance of the fish increased significantly after the training [37]. The mean critical speed of the swimmer group increased from 33.2 cm/s or 11.5 body lengths per second (bl/s) to 41.6 cm/s or 14.4 bl/s after 3 weeks (27% improvement, p = 0.00029) up to 43.9 cm/s or 15.2 bl/s after 5 weeks (36% improvement, p = 7.8e-8, n = 19). The control group did not improve significantly in 5 weeks, namely 34.2 cm/s to 36.0 cm/s or 11.7 bl/s to 12.3 bl/s during the final measurement (+6.9%, n = 20) (Fig 1).

Zebrafish continue to grow in adulthood; therefore, we measured the body length too. Body length of the swimmers increased slightly but significantly (+4.7% from 29.1±1.9 mm to 30.4±1.9 mm, n = 20 (19 after training), p = 0.018), while the control group showed no significant change (+0.7% from 29.4±1.8 mm to 29.6±1.5 mm, n = 20). Data from both males and females were used in these plots.

Exercise is associated with skeletal muscle hypertrophy and accordingly, we could show an increase in body weight of male swimmers from before to after the training period (+18.3% from 0.37±0.05 g to 0.44±0.06 g, n = 10 (n = 9 after training), p = 0.011). No significant weight change could be seen in the control group (+4.5% from 0.38±0.06 g to 0.39±0.04 g, n = 10), whereas the weight of the swimmer group compared to the control group after training showed a significant difference as well (p = 0.041) (Fig 1). Female group body weight was not taken into account due to the absence or presence of eggs in the body. We conclude that endurance training leads to an increase in body mass in adult zebrafish.

Interestingly, the behaviour of the fish adapted to the training too. During the first training day, fish were swimming in a rather nervous and unorganised way. They swam back and forth, discovered their new environment and had to get familiar with their new situation facing counter-current flow. During the training period, an improvement of swimming ability and technique was observed, the swimming style appeared smoother and there appeared to be less unnecessary movements. The so-called burst and glide swim style could be observed in all the groups during training [38]. Furthermore, we noticed behaviour that appears as if one fish plays the role of a group leader—a strong fish that stays behind all the others and pushes his colleagues that are falling behind (for both behavioural observations see S1 Video).

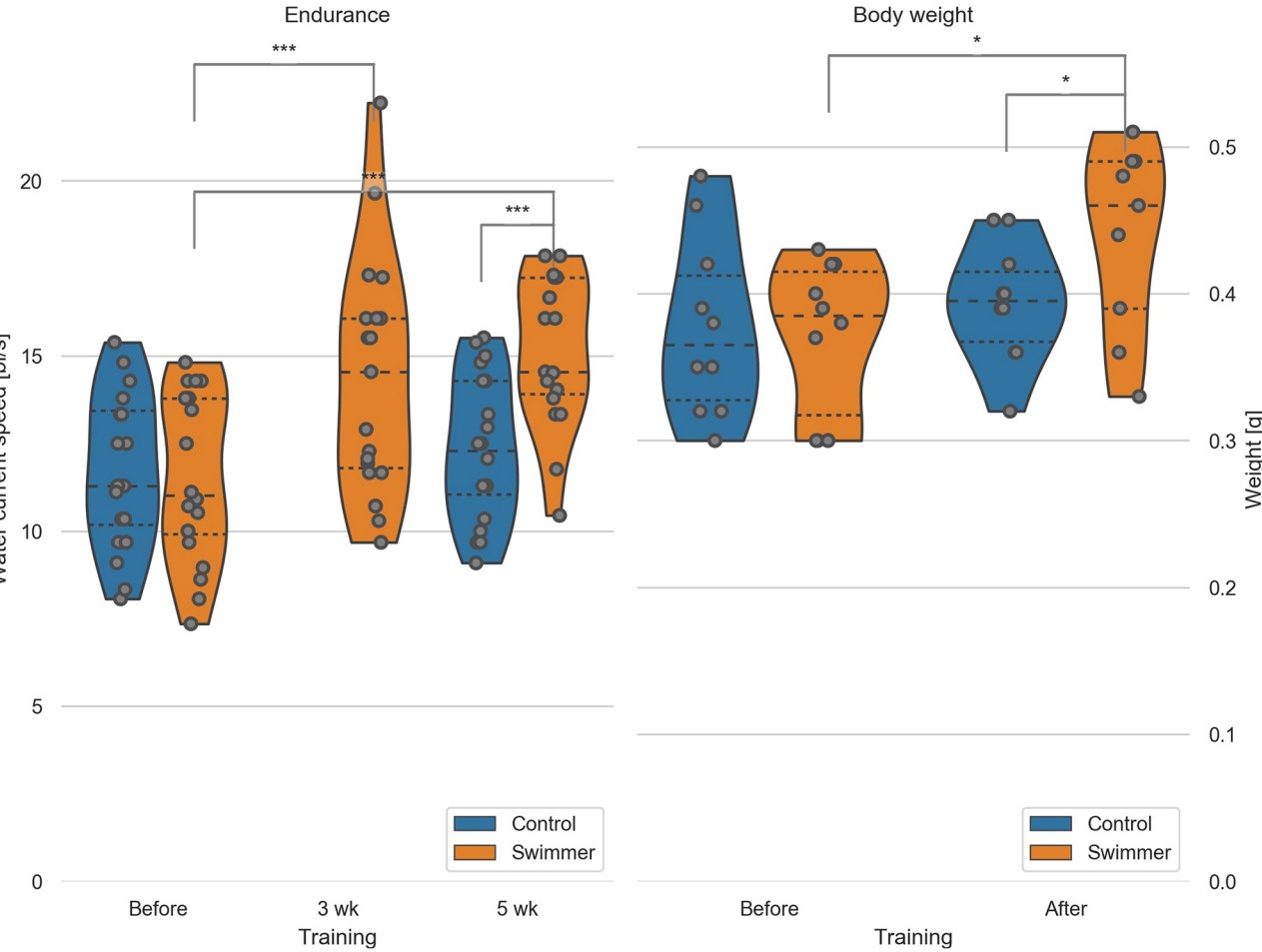

**Fig 1. Critical speed test of swimmer and control group before, during and after the training period; male fish body weight before and after training.** Left: max. achieved swimming speed by untrained and trained fish (n = 20 per group, except for swimmers at 5wk n = 19). Data for the control group at 3 weeks were not recorded. Significant differences are found for the swimmers in the performance before training compared to either 3 weeks (p = 0.00029) or 5 weeks of training (p = 7.8e-8). Controls and swimmers at 5 weeks also showed a significant difference in critical speed (p = 2.7e-6). All other combinations are not significant. Right: weight of all the fish was measured before and after 5 weeks of swimming (n = 10 for each group). Significant differences are found in the weight of the swimmers before and after training (p = 0.011) and between the control and swimmer group after training (p = 0.041). wk = week, *: p<0.05, ***: p<0.001, lines within the plots show the quartiles of the respective distributions.

## Normal gill morphology

The respiratory organs of the zebrafish, the gills, are located in two branchial chambers that lie on each side of the body, behind the eyes. These chambers are covered by an osseous lid, called operculum, which protects the gills from physical and mechanical damage (Fig 2). The gills consist of two elements: the arches and the filaments. In zebrafish, each branchial chamber contains four gill arches. They are made up of a bony skeleton and provide the structural support for the filaments. The filaments can further be divided into primary and secondary filaments (see also Fig 2C and 2D). Each primary filament consists of a cartilage pillar and an afferent and efferent blood vessel. The gas exchange happens in the secondary filaments that are attached to the primary filaments like little leaves to a twig. Within the secondary filaments lies a capillary network with its blood flow against the water current (counter-current flow), guaranteeing the maximally possible oxygen uptake.

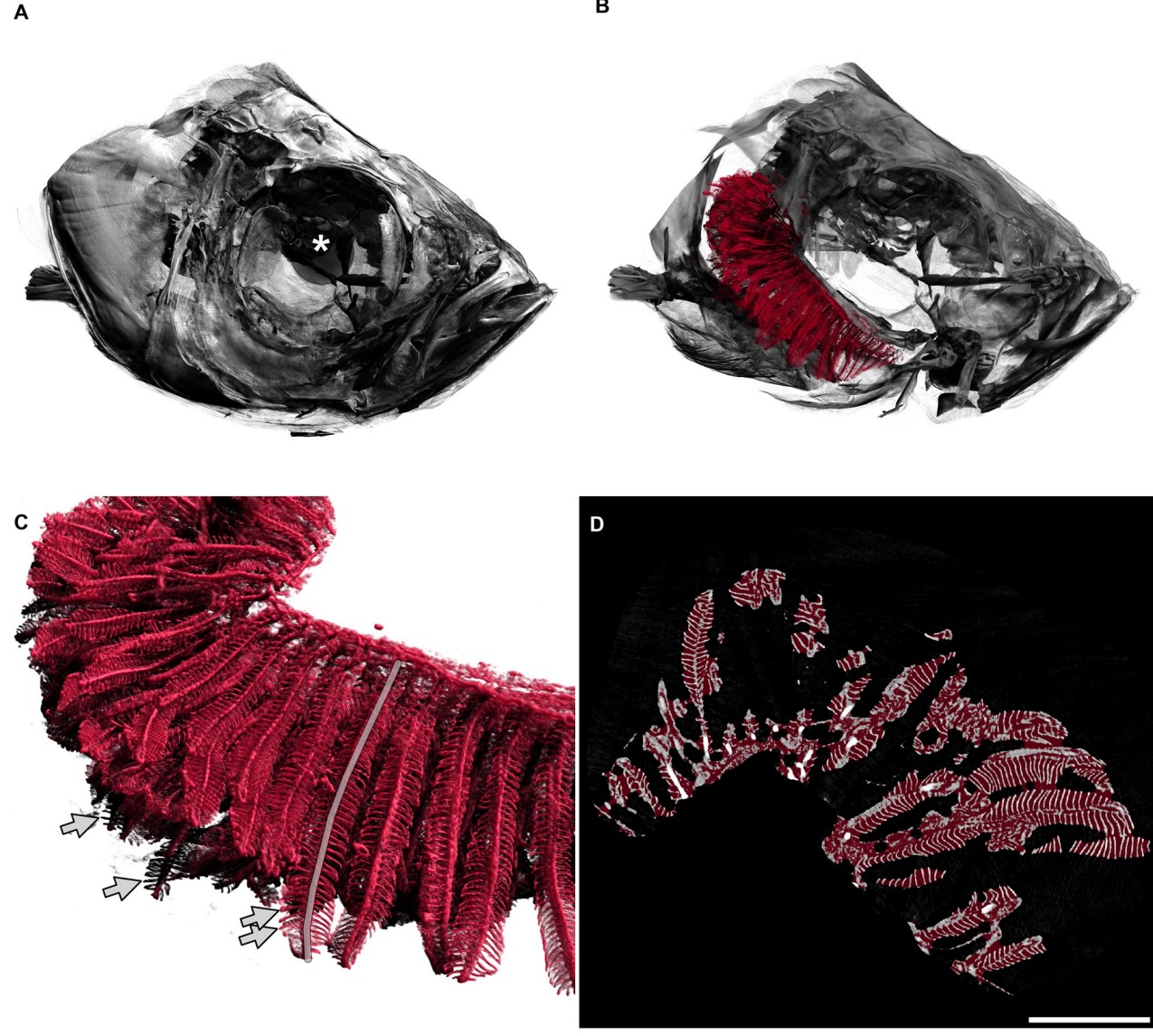

**Fig 2. 3D visualisation of a tomographic scan of a fish head from the control group.** A: Fish head. The diameter of the whole eye (centre marked with a white asterisk) is approximately 0.83 mm. B: The delineated gills in red are shown inside the head of the fish, operculum removed, only right arches of the gills are shown. The gill arches lie within the branchial chamber. In this image, primary filaments are mainly pointing to the left of the image (back of fish). C: Detailed view of gills. Secondary filaments are seen as leaf-like structures attached to the primary filaments. The semitransparent grey line marks one primary filament. Arrows mark the tips of four secondary filaments. D: Two-dimensional view of the gills, e.g. one slice of the tomographic data set where all three-dimensional measurements were based on. The red overlay denotes the estimation of the hull of the gill organ. The filling factor of the gills shown in the right panel of Fig 4 has been calculated by dividing the red volume by the white volume for each animal. Scale bar 0.5 mm. An animated version of the three-dimensional visualisation of the fish head can be found in S2 Video.

## Adaptation by increased filament number and length

We compared the morphology of separated gill arches (blinded printed photographs of arches 2 and 3 of both groups, 40 photos in total) regarding primary filament length and number of secondary filaments on primary filaments. The first visual impression was that the angle between primary and secondary filaments in swimmers was closer to the right angle compared to controls, meaning that the secondary filaments of swimmers pointed more to the sides whereas in controls they grew more towards the tip. As a consequence, the gills of the

swimmer group appeared less compact (with more room around the secondary filaments). Swimmers also appeared to have longer and more numerous secondary filaments, especially in the tip regions.

To support our first visual impression, we measured the length of the primary filaments and counted the number of secondary filaments in a clearly defined region of interest (longest 5 primary filaments of each arch 2 and 3). The mean length of primary filaments in trained fish was 6.1% higher in the swimmer group than in the control group (1639±228 $\mu$m versus 1545±148 $\mu$m, n = 10 fish per group with 5 filaments counted on both arch 2 and 3, therefore n = 100 per group, p = 0.00043). The mean number of secondary filaments per primary filament was significantly higher in the swimmer group versus the control group (+7.7%, 49±5 versus 46±3, n = 100 per group, p = 9e-9) (Fig 3). The distributing proportion of the secondary filaments within the upper and lower half of the primary filament is equal in swimmers and control (52.5% vs. 47.5%). For this measurement, the midpoint of the length of each primary filament was defined in Fiji and the secondary filaments were counted above and below this point.

As a conclusion of this semi-quantitative analysis, we suggest that exercise might induce gill growth in adult zebrafish.

## Adaptation by augmented gill volume

After the semi-quantitative analysis of a defined region of interest, we wanted to quantify the whole organ volume. After scanning the fish head with micro-CT, we were able to reconstruct the whole organ and to measure its volume as specified above.

The gill volume of trained fish was significantly larger than in control fish (+11.8%, 0.55 ±0.09 mm$^3$ versus 0.49±0.07 mm$^3$, n = 10 per group, p = 0.048) (Fig 4). We extrapolated the hull of the gills by filling the small voids between the secondary filaments with a closing filter [39]. This is analogous to covering the gills with cling-film and gives us an approximation of the total volume which the gills occupy in the animal. Dividing this hull volume by the gill volume calculated above gave us an estimate of the filling factor of the gills, e.g. the space-filling complexity. The gills of the swimmer group are filling significantly less space in the total organ hull (-8%, 35.9±2.0% for the swimmer versus 38.8±2.9% for the control group, p = 0.0088). The results confirm the qualitative finding from the SEM images and helps us to conclude that the gills of the swimmer group are less compact than the gills of the control group. We thus expect that the flow of oxygen-rich water is facilitated in the gills of the swimmers.

## Adaptation by increased oxygen consumption

We showed that gill volume is higher in exercised fish and that the filament morphology changes to support better exchange of oxygen and carbon dioxide. Next, we wanted to approach the more functional aspects of the gills. We measured the oxygen consumption during moderate swimming (training speed) of the fish before and after the training period of both controls and swimmers. Before training, the $O_2$ consumption did not differ significantly between the groups (controls: 0.033±0.011, swimmers: 0.034±0.013, 3.18% difference, n = 10 per group). The large variability of values was most likely due to different responses to the measuring chamber—signs of stress in some animals were accompanied by higher oxygen consumption.

After the training, the oxygen demand within swimmers increased slightly (+2.6% to 0.034 ±0.005, n = 9) and dropped in controls (-23.7% to 0.026±0.008, n = 10). Comparing the values after the training, the swimmers thus consume significantly more oxygen than the control fish (+30.4%, p = 0.0081) (Fig 5). The missing increase in swimmers oxygen consumption after

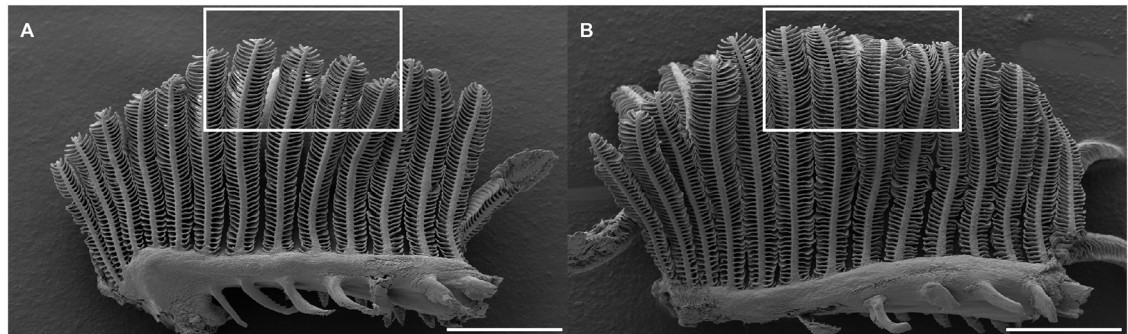

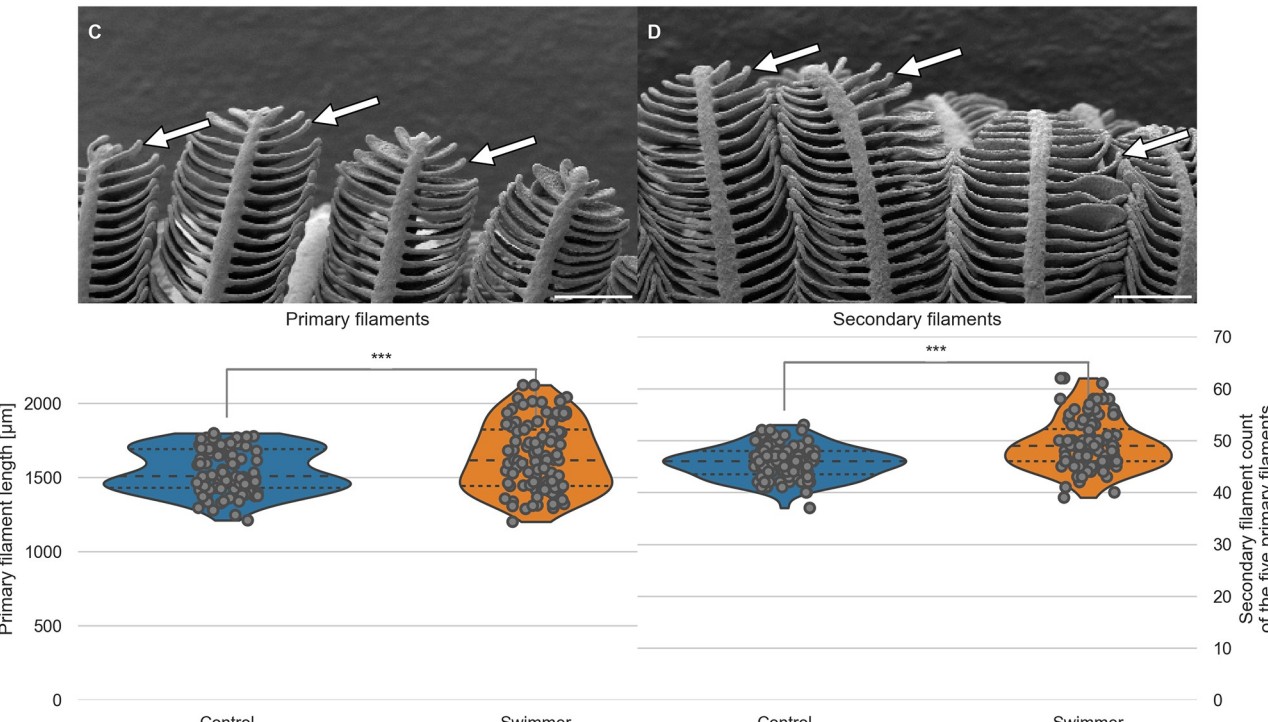

**Fig 3. SEM images of one separated gill arch; length of primary filaments and number of secondary filaments per primary filament.** A, B: Gill arch of control (A) and swimmer fish (B) with primary filaments pointing up vertically, secondary filaments are seen on each side of the primary filaments (examples shown with arrows). Also visible are the gill rakers, facing towards the pharynx and preventing food particles from exiting between the gill arches. After 5 weeks of training, SEM scans were printed and compared morphologically. The white frame marks the region of the zoom shown in panel C and D. Scale bars: 0.5 mm. C, D: Detailed view of the tips of the gill arches. Note the longer appearing secondary (arrows) of filaments of the swimmer (D) and their more horizontal appearance. Scale bars: 0.1 mm. Bottom row: Graphs from semi-quantitative measurement of primary filament length and number of secondary filaments on primary filaments, controls and swimmers after the training period. Arches 2 and 3 of each fish were taken into account (n = 100 per group). Left: The length of the five longest primary filaments increased significantly in trained fish (p = 0.00043). Right: secondary filament count on the 5 longest primary filaments of controls and swimmers, the swimmers showed a significantly higher number of secondary filaments (p = 9e-9). ***: p<0.001, lines within the plots show the quartiles of the respective distributions.

training can be explained with the fact that the fish of both groups displayed markedly less objective signs of stress (fast breathing, swimming with rapid directional changes) with equal measuring circumstances during the second measurement. These objective signs have already been described before [40], together with a decrease in cortisol levels in the blood [41].

## Adaptation by amplification of secondary filaments

After observing the obvious growth of the primary filament and the increase in number of secondary filaments and while studying the SEM pictures of swimmer gills, we came to the

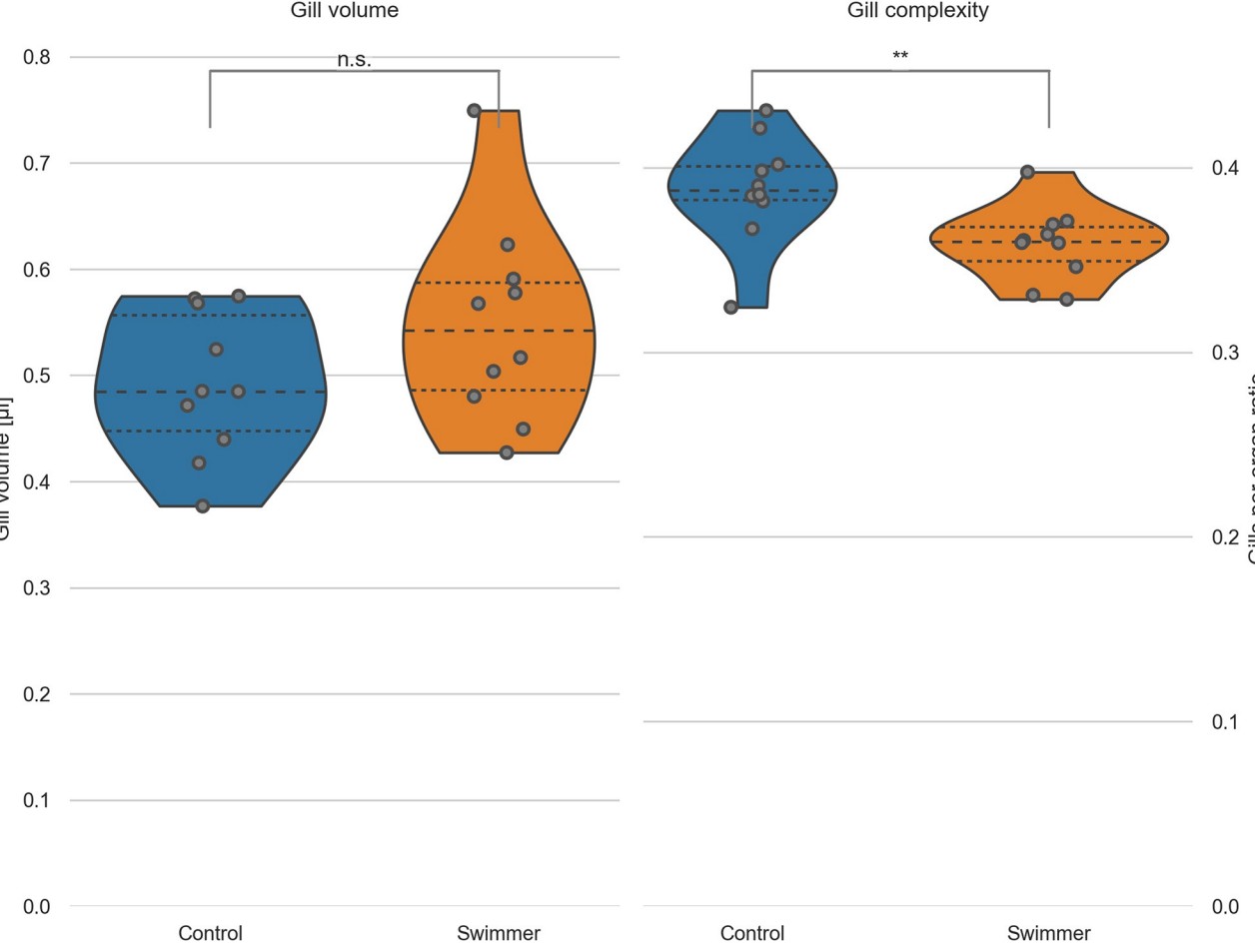

**Fig 4. Gill volume and filling factor of gills calculated from micro-CT data.** Left: the total volume of the gills was calculated from micro-tomographic assessment, after selecting a VOI and binarizing the image into gills and background. Data from controls and swimmers, showing a significant increase after 5 weeks of training (p = 0.048, n = 10 for each group). Right: Calculation of the ratio of gills per organ area (see explanation in the text). The swimmers have significantly less gills per organ, e.g. more room between the filaments (p = 0.0088, n = 10). *: p<0.05, **: p<0.01, lines within the plots show the quartiles of the respective distributions.

inference that the new budding takes place on the tip of the primary filament. Taking a closer look at the tips, it is obvious that they represent different stages of development. The very small secondary filaments are pointing more towards the tip whereas the longer ones are oriented more to the sides. The tip of the primary filament can appear more or less thick. Our hypothesis is, that first of all the tip of the primary filament thickens. Consecutively, at one side the tissue starts to separate from the tip into a secondary filament, similarly to the development of digits in embryo limb buds (Fig 6A to 6C). While the secondary filament grows, its orientation moves progressively sideways, like a flower opening its petals (Fig 6D and 6E). Finally, with additional growth in length, the new secondary filament joins the already existing ones on the side of the primary filament (Fig 6F). We propose to distinguish three main stages of secondary filament formation: The thickening, the sprouting stage and the growth.

## Adaptation by proliferation in budding secondary filaments

Now we discovered the ability of gills to increase their volume by growth of the primary filament and new buddying of secondary filaments and we proposed the stages of budding. But are those changes merely cell hypertrophy or does training stimulate proliferation rate?

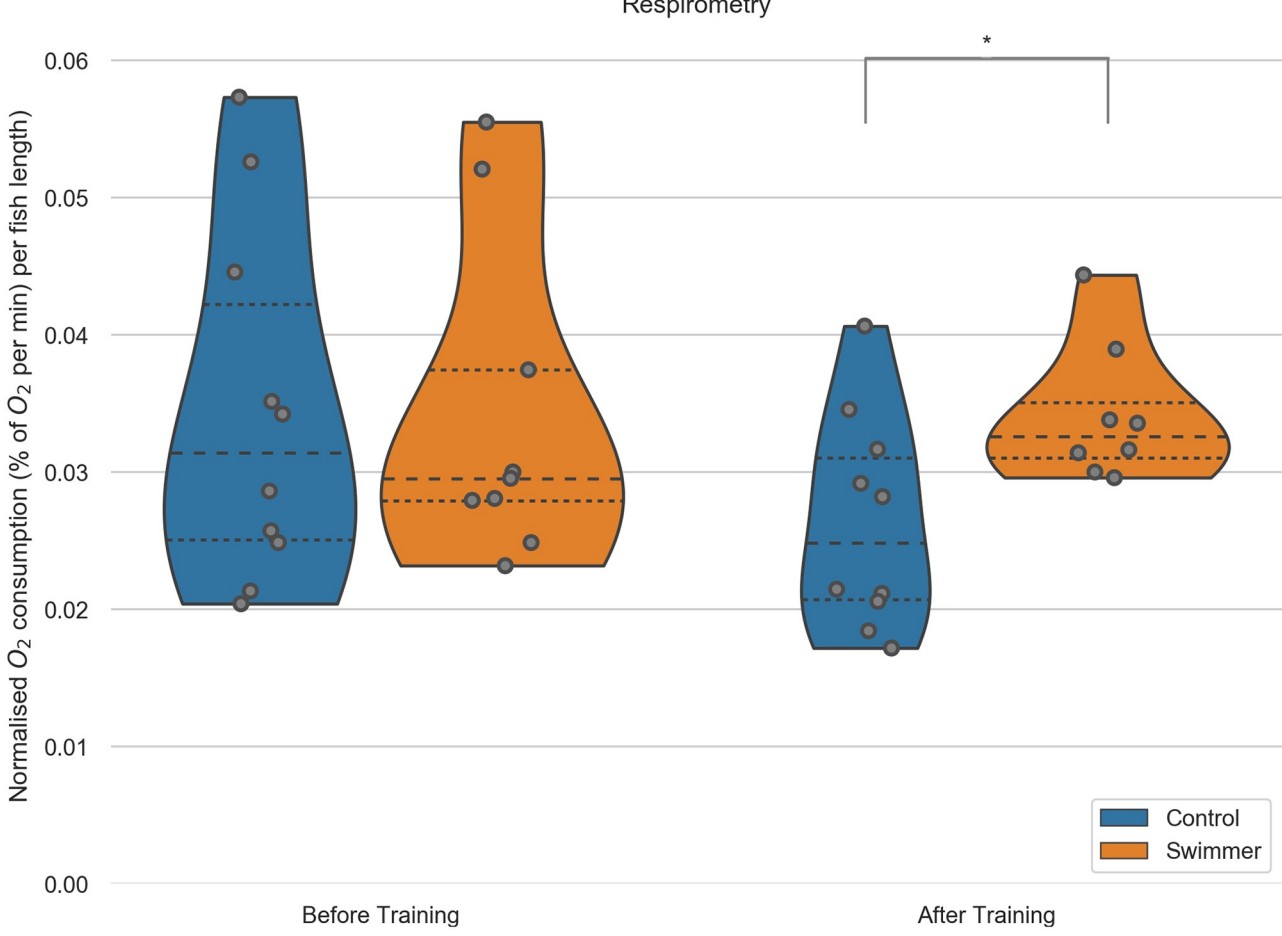

**Fig 5. $O_2$ consumption during moderate exercise (training speed) in swimmer and control fish.** Each fish was measured individually in a swim tunnel respirometer before and after the 5 week training period (n = 10 for each group before training, after training n = 10 in the control group, n = 9 for swimmers). Swimmers show a significantly increased oxygen consumption compared to the untrained control group after training (p = 0.0081). **: p<0.01, lines within the plots show the quartiles of the respective distributions.

To visualise nuclei that underwent division during the course of training, swimmers and controls were exposed to BrdU, which was then detected with antibody staining. Our SEM data indicated that the growth likely takes place in the tip regions, which is the reason why we decided to focus on the tips for our analysis. We first checked that the BrdU-positive spots are indeed newly divided nuclei by showing colocalisation with the DAPI staining. Since the amount of tissue per imaged region of interest was variable, we normalised the number of BrdU-positive nuclei to the total number of nuclei. This number was significantly higher in swimmers than in controls. At the tip of the gills, we measured an increase of 59.5% from 0.12 ±0.08 to 0.19±0.12 (p = 0.0074). At the base of the gills, we measured an increase of 98.6% from 0.07±0.06 to 0.15±0.04 (p = 0.00084). In 6 swimmers and 6 controls, cell division was counted on two gill arches per fish, two images were taken per arch) (Fig 7).

## Discussion

Endurance exercise leads to increased demands for oxygen and thus to an adaptation of the pathway of oxygen, consisting of the gas exchange organ, heart and blood, microvasculature,

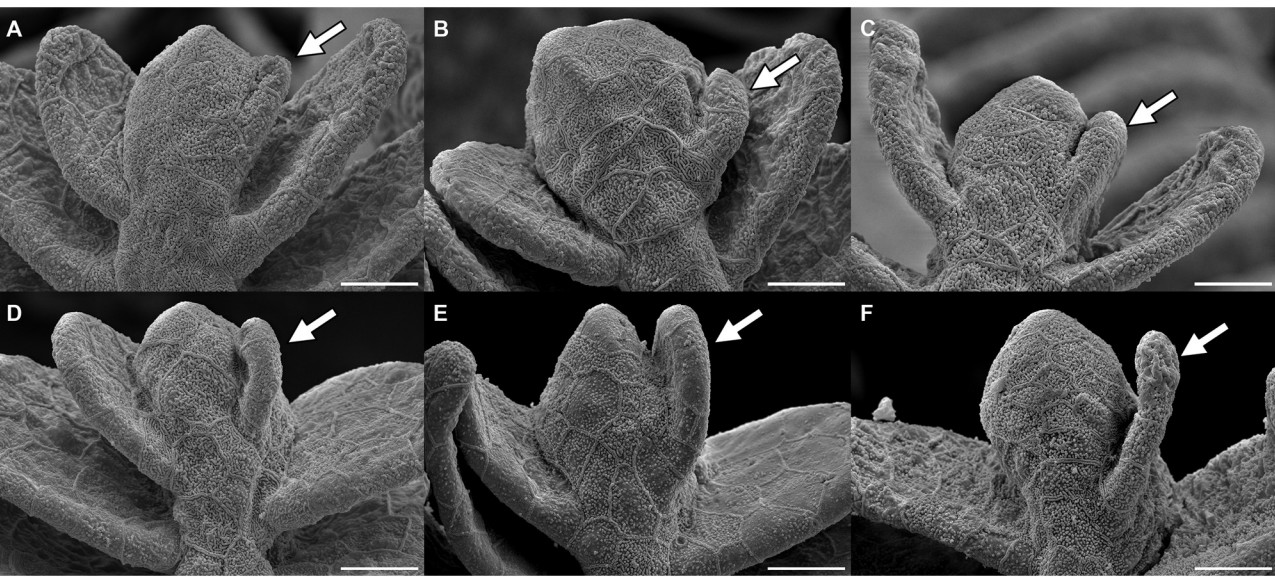

**Fig 6. Illustration of the different stages of secondary filament growth.** Read from A to F: the images show the sprouting stages of a new filament (arrow) on the right of the primary filament tip with initial thickening, progressive separation from the tip and growth of the newly developed filament. Scale bar: 15 $\mu$m.

and mitochondria [1]. Of these four components, adaptation of the latter three has been well established. However, the plasticity of the gas exchange organ remains controversial.

In our study, endurance training for 5 weeks and 6 hours per day led to a major improvement in performance (+36%), to an improvement of the swimming technique and a better organisation of the group. Throughout the training period, we observed improvements that are corresponding to the objective stress signs and group leader behaviour, mentioned in the results section and shown in S1 Video. Finally, the fish showed a more regular swimming pattern without major changing in speed and direction.

Maximal performance of zebrafish declines around 6% for every 10% progression in their lifespan [37]. The weaker performance of 12 bl/s in our elderly cohorts (age 18-23 months) compared to 18 bl/s reported for ca 2 months old zebrafish, was thus to be expected [26]. Optimisation of swimming patterns has been observed by others too: amplitude of tail beats increased with exercise [37]. This was linked to an increased propulsive force, which would help fish to maintain a stable position against the water current, and corresponds well to our qualitative observations.

The body mass of male fish increased significantly after the training (weight +18%, length +5%). This effect has been observed before in young zebrafish and proven to be due to muscle growth, which is a plausible explanation in our case, too [26]. The oxygen consumption increased as well: the trained group, swimming at a moderate speed, consumed 27% more oxygen than the control group, which we explain by the increased body mass of the swimmers. A positive correlation between body weight and oxygen consumption has already been shown in humans as well [42]. The swimmers used around 3% more oxygen than before the training, which would not *per se* be a relevant difference, but at the same time, the control group consumed 24% less oxygen during the second measurement than they did during the first one. We believe this drop of oxygen consumption in the control group to be due to the observed calmer behaviour of the fish the second time they were in the respirometry chamber: stress is known to be associated with an increased oxygen consumption [40, 41].

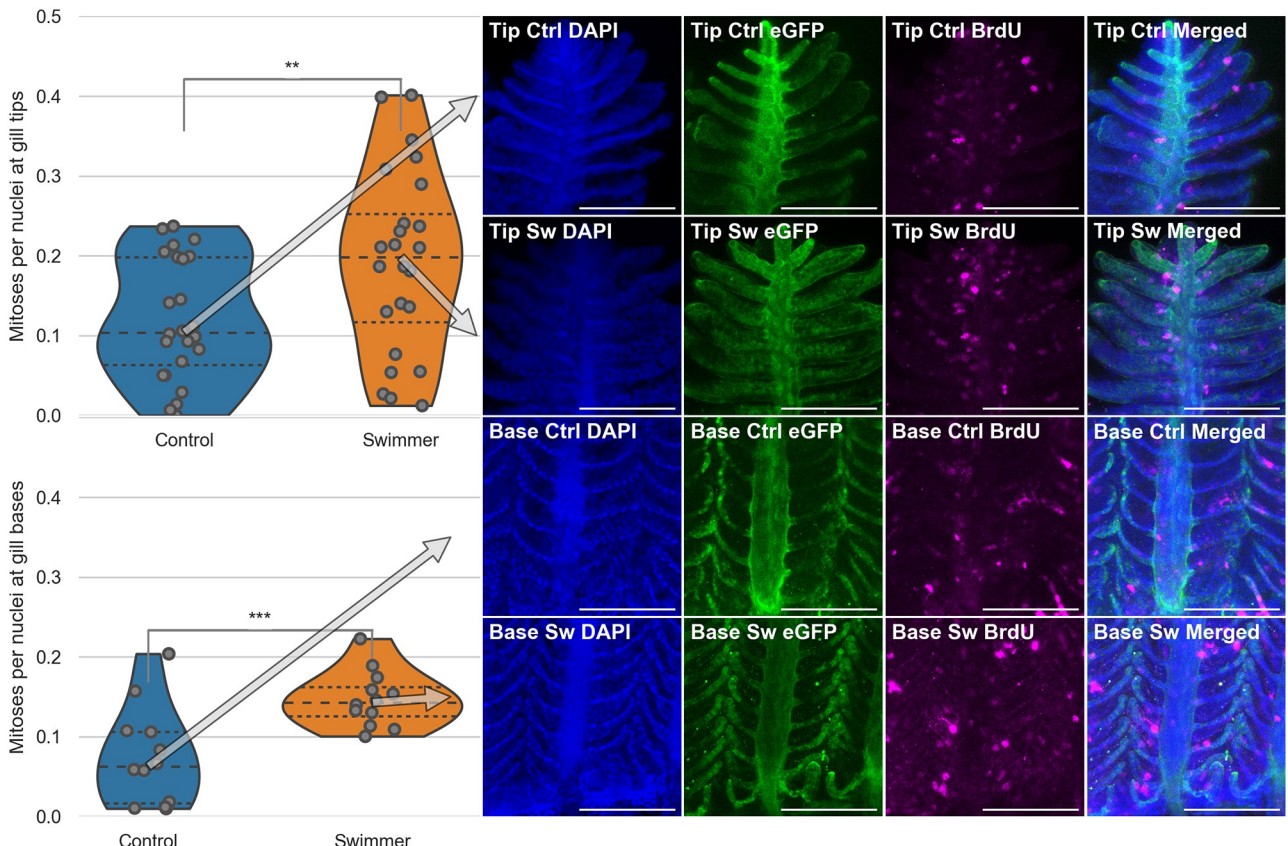

**Fig 7. Immunostaining of gill filaments and number of mitoses per total number of nuclei.** Top half: Data and images from the tips of the gills, bottom half: Data and images from the gill bases. Column 1: Plots of the number of mitoses per total number of nuclei in immunostained gill tips or bases (n = 6 for each group). After 3 weeks of training, the trained fish show a significantly higher number of dividing cells in their gills, compared to controls, both at the tips of the gills and at the base of them (tips: p = 0.0074, base p = 0.00084). The arrows point from the median value to the corresponding row of microscopy images. Columns 2-4: Staining of the nuclei in blue (DAPI), endothelium in green (eGFP) and mitoses in magenta (BrdU). Column 5: Composite image of the three channels. The microscopy images shown correspond to the median value of the data in the first column. Scale bar: 0.1 mm. **: p<0.01, lines within the plots show the quartiles of the respective distributions.

Micro-computed tomography of whole gills scanned *in situ* showed that they were significantly larger (+12%) than control gills, and that they were less compact (filling factor -8%), which facilitates the water flow through the organ and thus gas exchange. The unexpectedly high variability of the gill volumes was likely due to different distributions of the grey values among the scans and thus different threshold values for the volume calculation. The Otsu method was nevertheless preferred to calculate the threshold as an objective and reproducible method for data with bi-modal distribution, in contrast to manual thresholding. SEM images revealed that the primary filaments were longer (+6.1%) and the secondary filament per primary filament count was higher (+7.7%). These data together indicate a marked increase in the gas exchange surface.

Previous studies have shown that the gill surface of other fish species may increase when oxygen supply becomes temporarily or permanently limited. Permanent exposure to lower oxygen concentrations and different swimming behaviour is also responsible for a higher volume of the gill cavity and a greater respiratory surface area in fish living in the littoral benthic zone (close to the coast and the ground, with low oxygen concentration), compared to fish living in the open sea. This was revealed in two sympatric morphs of *Salvelinus alpinus* (Arctic charr) [43]. In *Carassius auratus* (Goldfish), hypoxia or endurance swimming induced a

marked increase in lamellar surface too (+71% after 48 hours under hypoxia, +43% after 48h of continuous swimming at 70% of the critical speed) [44]. Since oxygen solubility in water drops with higher temperatures, gill remodelling has also been observed in response to elevated water temperature [45, 46].

Detailed images of the morphology of the filament tips suggested that new secondary filaments might grow from the tips by a process we called filament budding. We expected to find more mitoses in filament tips of trained fish, and since we observed the steepest improvement of swimming performance during the first 3 weeks, we quantified mitotic events in this period. As expected, the percentage of nuclei of newly divided cells in gill tips marked by BrdU staining was significantly higher in swimmers (+60%). This is in line with the previously reported fast response of the gills to external stimuli that we stated above. The 48 hours of constant swimming Fu et al. performed, corresponds to 8 days of our training regime, i.e. half of the 3-week-protocol [44]. However, the authors consider these acclimation responses to be temporary and expect them to be reversed after the fish return to normal conditions. In contrast, our training regime resulted in gill enlargement that persisted at least several days after the endurance exercise has been stopped.

Opposite effect on gill mitotic index, i.e. a reduced proliferation rate and increased apoptosis, has been reported previously in *Carassius carassius* (Crucian carp) after exposure to hypoxia (14 days of 6-8% oxygen saturation) [47]. Although the adaptation led to +50% increased tolerance to hypoxia and documented changes in the lamellar surface exposed to water, strangely, the secondary lamellae were not visible at all in SEM micrographs of control fish. Another group reported hypoxia-induced gill surface modification [48]. Our data recapitulated neither of these phenomena. A possible explanation for this could be that severe hypoxia might induce a pathological phenotype while exercise has beneficial effects.

Endurance-training-related adaptations of the cardiovascular system and the internal oxygen transport of different vertebrate species are similar. We showed that the gas exchange is optimised in exercising fish: could it be that the adult mammalian lung may adapt too?

As stated in the introduction, swimming correlated with improved lung parameters in several studies [16–18] However, the mentioned differences have been reported in the growth phase of adolescents, and human lung continues to grow until adulthood [20]. Adaptation possibilities of adult lung tissue remain a controversially discussed topic. Several reasons might account for those divergent study results. Unlike gills, lungs are closed in the thoracic cavity and there simply might not be enough space for extra growth: re-initiation of growth of adult human lungs has so far only been proven after lung injury or disease with a loss of tissue, but not upon exercise. Another reason for missing adaptation of the lungs could be that the human pathway of oxygen (even in swimmers) has a different 'bottleneck' and that the diffusion capacity of the lung is simply not the limiting factor in the supply of $O_2$. Weibel et al. have already suggested this. They concluded that, due to its limited malleability, the lung needs to have excess capacity in order to allow changes in the subsequent steps of the respiratory pathway and to react to different $PO_2$ levels in the environmental air (e.g. in high altitude) [1].

Further research with mammalian models will be necessary to answer those questions and to gain a more profound insight into the adaptive changes possible in adult lungs.

## Conclusion

We present evidence of the long-lasting morphological adaptation of respiratory organ of adult animals to a physiological stimulus. Specifically, we measured an increase in primary filament length (+6.1%), number of secondary filaments per primary filament (+7.7%), and total gill volume (+11.8%) in adult zebrafish after endurance exercise. We proposed that gill

filaments may re-initiate their growth by a process we call 'gill filament budding'. We found probable stages of this process in SEM images and we proved an increased number of mitoses in gill filament tips, too (+60%). These morphological adaptations likely enabled better gas transfer: trained fish consumed more oxygen than controls when swimming at moderate speed (+30%), and the critical speed at which fish could swim increased by 36%. We noticed an increase in body mass too, in line with previous studies with zebrafish. Whether mammalian lung can regrow after exercise too, remains to be investigated.

## Supporting information

**S1 Fig. Quantification of BrdU+/eGFP+ cells with Imaris.** A: Endothelial surface in grey (based on relative eGFP channel intensity), showing the capillary network. Scale bar: 0.1 mm. B: Detailed view of square in A. Examples of BrdU positive endothelial cells ($^*$) and of BrdU-positive cells of other origin (o). Scale bar: 5 $\mu$m. C: Plots of the estimated surface at the base and tips (p = 0.00053).
(TIF)

**S2 Fig. Regression plots.** Left: Weight after training to filament count. $R^2$ controls 0.518 (p = 0.029, $^*$). $R^2$ swimmers 0.576 (p = 0.011, $^*$). Middle: Weight after training to normalized) gill volume. $R^2$ controls 0.042 (p = 0.57, n.s.). $R^2$ swimmers 0.071 (p = 0.46, n.s.). Right: Weight after training to gill complexity. $R^2$ controls 0.105 (p = 0.36, n.s.). $R^2$ swimmers 0.039 (p = 0.59, n.s.). The translucent bands mark the 95% confidence interval.
(TIF)

**S1 Video. Video of burst and glide swim style.**
(MP4)

**S2 Video. Three dimensional visualization of the tomographic scan of one of the zebrafish heads.**
(MP4)

## Acknowledgments

We thank the following people:

- Beat Haenni, Werner Graber, Jeannine Wagner-Zimmermann and Adolfo Odriozola for excellent sample preparation and imaging (SEM)

- Stefan Tschanz and Yury Belayev for valuable inputs about imaging and quantification

- Regula Buergy, Eveline Yao and Sara Soltermann for essential help in the fish facility

- Inês Marques, Xavier Langa and Alexander Uwe Ernst for help in sample preparation and method optimisation of the immunostaining

- Team from FIWI (Center for Fish and Wildlife Health, Vetsuisse Faculty, University of Bern) for kindly providing the material for respirometry

## Author Contributions

**Conceptualization:** Matthias Messerli, Dea Aaldijk, Helena Röss, Valentin Djonov.

**Data curation:** David Haberthür.

**Formal analysis:** Matthias Messerli, Dea Aaldijk, David Haberthür, Fluri A. M. Wieland.

**Investigation:** Matthias Messerli, Dea Aaldijk, David Haberthür, Helena Röss, Carolina García-Poyatos, Marcos Sande-Melón, Oleksiy-Zakhar Khoma, Fluri A. M. Wieland, Sarya Fark.

**Methodology:** Matthias Messerli, Dea Aaldijk, Helena Röss, Sarya Fark.

**Project administration:** Helena Röss, Valentin Djonov.

**Resources:** Valentin Djonov.

**Software:** David Haberthür.

**Supervision:** Valentin Djonov.

**Validation:** Matthias Messerli, Dea Aaldijk, David Haberthür, Helena Röss.

**Visualization:** David Haberthür.

**Writing – original draft:** Matthias Messerli, Dea Aaldijk, David Haberthür.

**Writing – review & editing:** Dea Aaldijk, David Haberthür, Helena Röss, Valentin Djonov.

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
