## [Decision Letter · Decision Letter 0]

23 Oct 2019

PONE-D-19-24766

Adaptation mechanism of the adult zebrafish respiratory organ to endurance training

p, li { white-space: pre-wrap; }

PLOS ONE

Dear Prof. Dr. med. Djonov,

Thank you for submitting your manuscript to PLOS ONE. After careful consideration, we feel that it has merit but does not fully meet PLOS ONE’s publication criteria as it currently stands. Therefore, we invite you to submit a revised version of the manuscript that addresses the points raised during the review process. Both reviewers are supportive of publication but have raised a number of points for your consideration. I think it is especially important to clarify the results of the BrDU study as both reviewers had questions about this aspect of the work. I also think some additional analysis of your data (as suggested by reviewer 2) is warranted, as is response to the request for clarification and additional citation from reviewer 1. The additional experiments suggested by reviewer 2 are an interesting suggestion that I hope you will consider. However, I do not consider them to be necessary for acceptance of a revised manuscript.

We would appreciate receiving your revised manuscript by Dec 07 2019 11:59PM. To enhance the reproducibility of your results, we recommend that if applicable you deposit your laboratory protocols in protocols.io, where a protocol can be assigned its own identifier (DOI) such that it can be cited independently in the future. For instructions see: http://journals.plos.org/plosone/s/submission-guidelines#loc-laboratory-protocols

We look forward to receiving your revised manuscript.

Kind regards,

Eric A Shelden, Ph.D.

Academic Editor

PLOS ONE

**Journal Requirements:**

**Comments to the Author**

1. Is the manuscript technically sound, and do the data support the conclusions?

Reviewer #1: Yes

Reviewer #2: Yes

2. Has the statistical analysis been performed appropriately and rigorously? 

Reviewer #1: Yes

Reviewer #2: Yes

3. Have the authors made all data underlying the findings in their manuscript fully available?

Reviewer #1: Yes

Reviewer #2: Yes

4. Is the manuscript presented in an intelligible fashion and written in standard English?

Reviewer #1: Yes

Reviewer #2: Yes

5. Review Comments to the Author

Reviewer #1: In this manuscript, Messerli et al. report on the effects of exercise by swimming on the respiratory performance of adult zebrafish. The authors show that swimming training increases swimming performance as determined by maximum swimming speed and that this response is accompanied by physiological responses consequent with increases respiratory performance. Namely, the authors show that swimming training increases oxygen consumption and the growth of the gills in terms of increased gill volume, length of primary filaments and number of secondary filaments per primary filament. Finally, the authors show that swimming training increases the number of proliferating cells in the gills, suggesting a possible mechanism for the observed increased gill growth. The experiments are well conducted, following previous established swimming training conditions for adult zebrafish, use appropriate sample sizes and choose relevant methodological approaches to investigate physiological responses to swimming training. Therefore, the results are meaningful and make this study a valuable contribution to the field of exercise physiology. This reviewer recommends that the authors take the following several aspects into consideration:

1. The authors need to clarify aspects of the swimming training and respirometry experiments. First, details of the swim tunnels used in this study need to be provided (volume, model number). Second, a clear indication of whether the swimming training was conducted on individual fish or on groups of fish (if this was the case, please indicate exactly the number of fish in the swimming chamber) is needed. Third, please clearly state where the non-swimming fish were housed during the experiments. Fourth, please indicate the value of the training speed in body lengths per second. Fifth, please clearly indicate whether oxygen consumption was determined on individual fish in the Respirometry section of M&M.

2. A justification for the use of transgenic zebrafish is needed and related additional analyses are strongly recommended (point 3). First, the authors should confirm whether the fli transgenic line use corresponds to fli11a:eGFP, as indicated in the manuscript (line 119), or instead to the fli1a:eGFP line that is commonly used as a vascular endothelial cell marker. If the latter, please correct. Second, does the reference provided indicate the initial description of this line or the research group that provided the line? If the former, please indicate the origin of the line used in these experiments.

3. With regards to the use of this transgenic line, it is not clear why the authors did not exploit the GFP labeling of endothelial cells to determine if proliferating (BrdU+) cells are indeed endothelial cells or not. This reviewer strongly recommends the authors to reanalyze the immunofluorescence data and provide quantification of BrdU+/GFP+ (i.e. proliferating endothelial cells) and BrdU+/GFP- cells (i.e. proliferating non-endothelial cells) so that the possible nature of the proliferating cells can be discussed.

4. The authors conclude that their data represent the first evidence of the morphological adaptation of a respiratory organ to a physiological stimulus. This statement is incorrect (Fu et al. show increased lamellar surface area in response to swimming training) and should be removed or reworded to indicate that the authors provide further evidence to support it.

5. Line 376. The authors should state more clearly what they imply by “this phenomenon”. Are the authors referring to the stress-reducing effects of swimming?

6. Lines 415-416. The description of the behavioral effects of endurance training is poorly worded. This reviewer suggests to better describe the behavioral effects in terms of schooling.

7. Line 468. Common names are not italized.

Reviewer #2: This is an interesting study and well written manuscript. OVerall, the authors appear to have conducted their studies carefully. I have only minor concerns.

First, the authors state that BrDU staining is primarily located at the tip of gill filaments in trained fish, but it is unclear from Figure 7 what the authors mean by this. The stained cells appear to be distributed throughout the region shown in the figure. If the authors are arguing that the whole region shown in Figure 7 represents the “tip” of the gill fiber, then other images showing more distal regions of the gill should also be included for comparison. Alternatively, the images shown do not clearly look like the mitotic figures are concentrated toward the tips of the structures shown, and clarification of their conclusion and/or more images to support their conclusion, might be warranted.

A second question concerns the data for mitotic index (Figure 7) and filament growth (Figure 3). While I accept that the average values for each group show significant increases in the trained fish, it certainly looks like many fish showed no changes versus controls. This should be commented on. In particular, I wonder if the authors can provide some data on which fish showed the best growth and mitotic index. For example, are these data correlated with overall growth of the animal? Could the authors not conduct regression analysis to show which variables do/do not correlate with gill growth?

Finally, assuming that the authors are willing to conduct additional experimentation, it seems like the issue raised above could be addressed better if images of the gills of individual fish could be obtained at intervals during the course of the training period. If such images could be obtained at a resolution sufficient to visualize primary or even secondary gill filaments, this might provide still more compelling data with regard to the manner and location of filament growth.

6. PLOS authors have the option to publish the peer review history of their article (what does this mean?). If published, this will include your full peer review and any attached files.

Reviewer #1: No

Reviewer #2: No

---

## [Author Response · Author response to Decision Letter 0]

12 Dec 2019

Reviewer #1

1. The authors need to clarify aspects of the swimming training and respirometry experiments.

First, details of the swim tunnels used in this study need to be provided (volume, model number).

Details on the two used swim tunnels (Model #SW10100 for training and model # SW10000 for respirometry) have been added to the manuscript, including direct links to the relevant pages on the Loligo website.

Second, a clear indication of whether the swimming training was conducted on individual fish or on groups of fish (if this was the case, please indicate exactly the number of fish in the swimming chamber) is needed.

Swimming training was performed in 10/group. Maximum swimming capacity was assessed in individual fish. We updated the Materials and methods section in the manuscript accordingly.

Third, please clearly state where the non-swimming fish were housed during the experiments.

We specified details on general housing of the fish and where the control fish were kept (in their regular tanks) in the Materials and methods section.

Fourth, please indicate the value of the training speed in body lengths per second.

Except for the speed increase of the swim tunnel on line 138 all speeds were given in cm/s and body length per second. On line 138 we would like to leave the value since this is a machine setting and we think "increase of 0.5 cm/s every two minutes" is more illustrative than "increase of 1.71 bl/s every two minutes for body lengths measured before training". It is unclear if we should remove the cm/s altogether, please let us know if this is the case.

The y-axis value of the left part of figure 1 was changed to show the speed in bl/s instead of cm/s.

2. A justification for the use of transgenic zebrafish is needed and related additional analyses are strongly recommended (point 3).

The presented study is just one branch of a broader project on exercise-induced angiogenesis in skeletal muscle. The fli1a:eGFP line was thus chosen to label endothelium. With respect to the 3R-initiative (replacement, reduction, and refinement) for the ethical use and the reduction of the number of animals sacrificed for science, we've looked at other exercise-induced adaptations and noticed interesting changes in gills, which we present in the manuscript. A short sentence on the choice of transgenic line was added under the Animals section in the manuscript.

First, the authors should confirm whether the fli transgenic line use corresponds to fli11a:eGFP, as indicated in the manuscript (line 119), or instead to the fli1a:eGFP line that is commonly used as a vascular endothelial cell marker. If the latter, please correct

This was indeed an oversight from our side and has been corrected in the manuscript. Thanks for pointing this out to us.

Second, does the reference provided indicate the initial description of this line or the research group that provided the line? If the former, please indicate the origin of the line used in these experiments.

The reference indeed provides only an initial description of the zebrafish line. The fishes were originally procured from the Zebrafish International Resource Center. The text has been updated accordingly.

3. With regards to the use of this transgenic line, it is not clear why the authors did not exploit the GFP labeling of endothelial cells to determine if proliferating (BrdU+) cells are indeed endothelial cells or not. This reviewer strongly recommends the authors to reanalyze the immunofluorescence data and provide quantification of BrdU+/GFP+ (i.e. proliferating endothelial cells) and BrdU+/GFP- cells (i.e. proliferating non-endothelial cells) so that the possible nature of the proliferating cells can be discussed.  

As mentioned above, we were not intending to quantify the proliferating endothelial cells in the gills. However, it is possible to construct an eGFP+ surface in Imaris software and look whether the BRDU-positive spots lie within (see Fig 1, panels A and B). This figure has been added as supplementary material to the manuscript.

Exact calculation of the endothelial surface is not trivial because of the unexpectedly high fluctuations in eGFP expression between fish. Uniform thresholding parameters (relative changes in eGFP intensities) inevitably cause over or underestimation of the endothelial surface in some samples.

Nonetheless, our calculations provide evidence that the endothelial surface in filament tips of trained fish is significantly larger than in controls (+21.8 %,  p-value 0.00053, n=12+12), while no changes in endothelial surface at the filament base was observed (p=0.16).

The relative difference between swimmers-tips and controls-tips was observed independently on the thresholding parameters and we are thus confident that there is a real difference. However, the generated surface is just an *estimation*, e.g. *not* identical with the actual surface of the endothelium, and thus given in 'arbitrary units'.

4. The authors conclude that their data represent the first evidence of the morphological adaptation of a respiratory organ to a physiological stimulus. This statement is incorrect (Fu et al. show increased lamellar surface area in response to swimming training) and should be removed or reworded to indicate that the authors provide further evidence to support it.

Our formulation was improper; we apologize. We cite the work of Fu et al. for the increase of lamellar surface but while they focused on the acute (and possibly transient, as the authors explain) adaptation to exercise, we've investigated more permanent effects that last at least for several days after the cessation of exercise. We aren't aware of any other publication on chronic (permanent) changes of gills after endurance training, but we've taken the formulation 'first evidence' out of the text to avoid misunderstandings, and clarified the differences between these two exercise models (48h without a break versus our 3- or 5-week regime) in the text.

5. Line 376. The authors should state more clearly what they imply by “this phenomenon”. Are the authors referring to the stress-reducing effects of swimming?  

We clarified the references to the objective stress-reduction signs.

6. Lines 415-416. The description of the behavioral effects of endurance training is poorly worded. This reviewer suggests to better describe the behavioral effects in terms of schooling.

We updated and expanded the explanation of the observation of reduced objective signs of stress and linked to the movie about the behavioral effects already mentioned in the results section.

7. Line 468. Common names are not italicized.

We have removed the italicization of Crucian carp on this line and from several other common names in the manuscript.

Reviewer #2

First, the authors state that BrDU staining is primarily located at the tip of gill filaments in trained fish, but it is unclear from Figure 7 what the authors mean by this. The stained cells appear to be distributed throughout the region shown in the figure. If the authors are arguing that the whole region shown in Figure 7 represents the “tip” of the gill fiber, then other images showing more distal regions of the gill should also be included for comparison. Alternatively, the images shown do not clearly look like the mitotic figures are concentrated toward the tips of the structures shown, and clarification of their conclusion and/or more images to support their conclusion, might be warranted.

We have added representative images from both the tip and base region of the gills to Fig07. 'Representative' means that the images correspond to the median value of the fishes in this row. Fig07 has been updated accordingly.

Since this image is now more dense, we decreased the font size of the image label. To be consistent, the font size of all the labels on all the figures has been decreased to the same size.

A second question concerns the data for mitotic index (Figure 7) and filament growth (Figure 3). While I accept that the average values for each group show significant increases in the trained fish, it certainly looks like many fish showed no changes versus controls. This should be commented on. In particular, I wonder if the authors can provide some data on which fish showed the best growth and mitotic index. For example, are these data correlated with the overall growth of the animal? Could the authors not conduct regression analysis to show which variables do/do not correlate with gill growth?

The criteria for the quantification of mitoses was based on threshold values for intensities in BrdU and DAPI channels as well as spot size and geometry. Analysis was done in a batch process for all scans. Despite cautious and strictly uniform staining and imaging of the samples, our results show a large variability within the groups. Repeating the batch analysis with different selection criteria did not affect the ratio of the values between the groups; we're thus convinced that we see a true effect. 

As we didn't label nor train the fish individually, we are not able to calculate the growth or weight gain of the individual fish, we could only give an average of the whole group. This measurement was never a part of our study setting and animal experiment permit. We thus only have pooled data for the whole cohort.

We plotted the regression of the different assessed parameters of the fishes. But apart from the obvious trends (fish weight and filament count or fish weight and gill volume), no significant or meaningful correlation was observed (as seen in Figures 3 to 6 below). Additionally, comparing the measured gill parameters prior and after training is not possible due to ex-vivo (destructive) character of the investigation. Please also see the answer to the next suggestion of yours for reasons to not perform more experiments.

A combined figure of the regression plots below has been added to the manuscript as a supplementary figure.

Finally, assuming that the authors are willing to conduct additional experimentation, it seems like the issue raised above could be addressed better if images of the gills of individual fish could be obtained at intervals during the course of the training period. If such images could be obtained at a resolution sufficient to visualize primary or even secondary gill filaments, this might provide still more compelling data with regard to the manner and location of filament growth.

Due to the ex vivo imaging approach in this study, we cannot assess the gill parameters of individual fishes in a longitudinal study. The post-mortem method makes this impossible, we can only study the gill parameters at the study endpoint. In vivo tomographic imaging of fishes is possible but lacks the resolution needed to distinguish the secondary filaments.

One possibility would be to could greatly increase the cohort size and remove a sufficient amount of fishes per timepoint to be studied (N>5) from the experiment for performing tomographic (and EM) imaging. This would then allow for assessing the data (pooled over the whole cohort) during the training period. Due to constraints in time, manpower and animal permit, we did not perform this in the present manuscript, albeit we fully agree that this would most probably provide still more compelling d

---

## [Decision Letter · Decision Letter 1]

14 Jan 2020

Adaptation mechanism of the adult zebrafish respiratory organ to endurance training

PONE-D-19-24766R1

Dear Dr. Djonov,

We are pleased to inform you that your manuscript has been judged scientifically suitable for publication and will be formally accepted for publication once it complies with all outstanding technical requirements.

With kind regards,

Eric A Shelden, Ph.D.

Academic Editor

PLOS ONE

Additional Editor Comments (optional):

Reviewers' comments:

Reviewer's Responses to Questions

**Comments to the Author**

1. If the authors have adequately addressed your comments raised in a previous round of review and you feel that this manuscript is now acceptable for publication, you may indicate that here to bypass the “Comments to the Author” section, enter your conflict of interest statement in the “Confidential to Editor” section, and submit your "Accept" recommendation.

Reviewer #1: All comments have been addressed

Reviewer #2: All comments have been addressed

2. Is the manuscript technically sound, and do the data support the conclusions?

Reviewer #1: Yes

Reviewer #2: Yes

3. Has the statistical analysis been performed appropriately and rigorously? 

Reviewer #1: Yes

Reviewer #2: Yes

4. Have the authors made all data underlying the findings in their manuscript fully available?

Reviewer #1: Yes

Reviewer #2: Yes

5. Is the manuscript presented in an intelligible fashion and written in standard English?

Reviewer #1: Yes

Reviewer #2: Yes

6. Review Comments to the Author

Reviewer #1: (No Response)

Reviewer #2: I believe the authors have adequately addressed the concerns that I raised in the previous round of review, and recommend acceptance.

7. PLOS authors have the option to publish the peer review history of their article (what does this mean?). If published, this will include your full peer review and any attached files.

Reviewer #1: No

Reviewer #2: No

---

## [Editor Report · Acceptance letter]

24 Jan 2020

PONE-D-19-24766R1 

Adaptation mechanism of the adult zebrafish respiratory organ to endurance training 

Dear Dr. Djonov:

I am pleased to inform you that your manuscript has been deemed suitable for publication in PLOS ONE. Congratulations! Your manuscript is now with our production department. 

With kind regards,

on behalf of

Dr. Eric A Shelden 

Academic Editor

PLOS ONE